# Quantifying the Impact of COVID-19 "Lockdown" on Physical Activity in Children and Adults with Implanted Cardiac Rhythm Devices: A Single Center Experience Using Cardiac Rhythm Device Accelerometer Data

Rebecca Fisher, David Jevotovsky [ORCID], Jessica Raviv and Barry Love *

Departments of Internal Medicine and Cardiology, Icahn School of Medicine at Mount Sinai, New York, NY 10029, USA
* Correspondence: barry.love@mssm.edu

**Abstract:** Background: In response to the COVID-19 pandemic, the US states of New York and New Jersey instituted a "lockdown" that closed schools and most businesses except for essential services. The public was urged to remain at home. The 78-day lockdown period extended from 22 March 2020 to 6 June 2020. We sought to evaluate the impact of COVID-19 lockdown on continuously recorded physical activity in our patients with congenital heart disease with implanted cardiac rhythm management (CRM) devices—pacemakers (PPM), defibrillators (ICD) and implantable loop recorders (ILR) enabled with accelerometers that translate motion into a measure of active hours/day. Methods: All patients from our congenital CRM database with accelerometer-enabled devices who had stable cardiac status residing in NY and NJ were included. Forty-one patients met the inclusion criteria; the median age was 29.6 years (range 7–60); 54% female; 23 ICD, 15 PPM and 3 ILR. The patient activity was averaged for the 2 months prior to lockdown, during the lockdown, and the 2 months afterward. Paired *t*-tests were used to compare activity before and during lockdown, and before and after lockdown. Each patient served as his/her own control. Results: Active hours/day decreased by a mean of 32% (±27%) from baseline ($p < 0.001$). A total of 32 patients experienced a decline, 6 had no change and 3 had an increase. Active hours rebounded after lockdown and were unchanged compared to pre-lockdown levels. Conclusions: The COVID-19 lockdown in NY/NJ during the Spring of 2020 resulted in a significant drop in active hours/day in children and adults with congenital heart disease. Active hours/day rebounded to baseline after restrictions were lifted.

**Keywords:** COVID-19; congenital heart disease; pacemaker; implantable defibrillator

## 1. Introduction

In response to the COVID-19 global pandemic, the US states of New York and New Jersey instituted a formal lockdown policy on 22 March 2020 [1]. People stayed home as non-essential businesses were closed statewide, and work and school shifted to remote settings. Public health announcements urged people to stay home, but many did not comment on how to maintain physical activity. The lockdown ended as Phase 1 of reopening started on 8 June 2020, [2] but many did not return to work or school, and significant restrictions remained on peoples' out-of-home activities.

It is well-understood that physical activity level is an important predictor of cardiovascular outcomes [3]. Therefore, the decrease in physical activity during the COVID-19 pandemic has been linked to an associated increase in the global burden of cardiovascular disease [4].

Several studies support the notion of decreased activity through analysis of self-reported questionnaires, interviews and smartphone data [5–10]. However, the association between lockdown and physical activity remains incompletely characterized from an objective standpoint. A highly accurate, continuous method to document physical activity

is to use the data from accelerometers in many enabled cardiac rhythm management devices (pacemakers, implantable defibrillators and implantable loop recorders) to track and record physical activity reliably [11]. Unlike other methods, such as smartphones, the data is continuously recorded in these devices and is not subject to patient compliance. Furthermore, the data resides in the devices for the continuous timeframe prior to, during and after the lockdown period.

This study sought to examine the effect of the public health lockdown on physical activity in a group of patients with cardiac rhythm devices.

## 2. Materials and Methods

Participants with implantable cardioverter defibrillators (ICD), permanent pacemakers (PPM) and implantable loop recorders were included in the congenital cardiac rhythm management (CRM) database. Those with accelerometer-enabled devices and stable cardiac status who resided in New York and New Jersey were included in this analysis.

Patient activity was measured by device activity reports in hours of activity per week. Activity was calculated using accelerometer information. Accelerometers utilize piezoelectric crystal sensors that detect changes in the frequency and amplitude of body motion as the patient moves. The sensors then generate an electrical signal that is proportional to patient movement. Company-specific algorithms translate the accelerometer data into a measure of physical activity and store the data in device memory which can be downloaded without wireless internet during routine clinic visits or via remote monitoring [11]. An "active hour" is considered approximately 60–70 steps/minute during that hour. The activity is reported by the devices as hours/day of activity.

Patient activity was averaged for the 2 months prior to lockdown, during the 78-day lockdown period, and the 2 months afterward. Paired T-tests compared activity before and during lockdown, and before and after lockdown. Each patient served as his/her own control.

## 3. Results

Participants were included from the congenital CRM database (*n*= 92). Those with accelerometer-enabled devices (44/92) and stable cardiac status who resided in New York and New Jersey were included. Forty-one met the inclusion criteria. Participants had a median age of 29.6 years (range 7–60), 46% were male, 23 patients had an implantable cardioverter defibrillator (ICD), 15 had a permanent pacemaker (PPM) and 3 had an implantable loop recorder (ILR) (Table 1). CRM devices were manufactured by Medtronic (*n* = 38) and Boston Scientific (*n* = 3).

**Table 1.** Demographic of study population.

| Age | Median 29.6 Years (7–60) |
| --- | --- |
| Gender (male) | 46% |
| ICD | 23 |
| PPM | 15 |
| ILR | 3 |

Abbreviations: ICD, implantable cardioverter defibrillators; PPM, permanent pacemakers; ILR, implantable loop recorders.

Among the 41 patients, active hours per day decreased by 32% ($\pm$ 27%) from baseline (4.05 $\pm$ 1.68 h/day) to lockdown (2.71 $\pm$ 1.59 h/day) ($p < 0.001$) (DF 37, T 2.03) (Figure 1). A total of 32 patients experienced a decline in activity, 6 had no change and 3 increased. Active hours rebounded by 53% ($\pm$50%) after lockdown (3.78 $\pm$ 1.48 h/day) and were unchanged compared to pre-lockdown levels ($p$ = NS). There was a 94.78% return to baseline activity when comparing before to after.

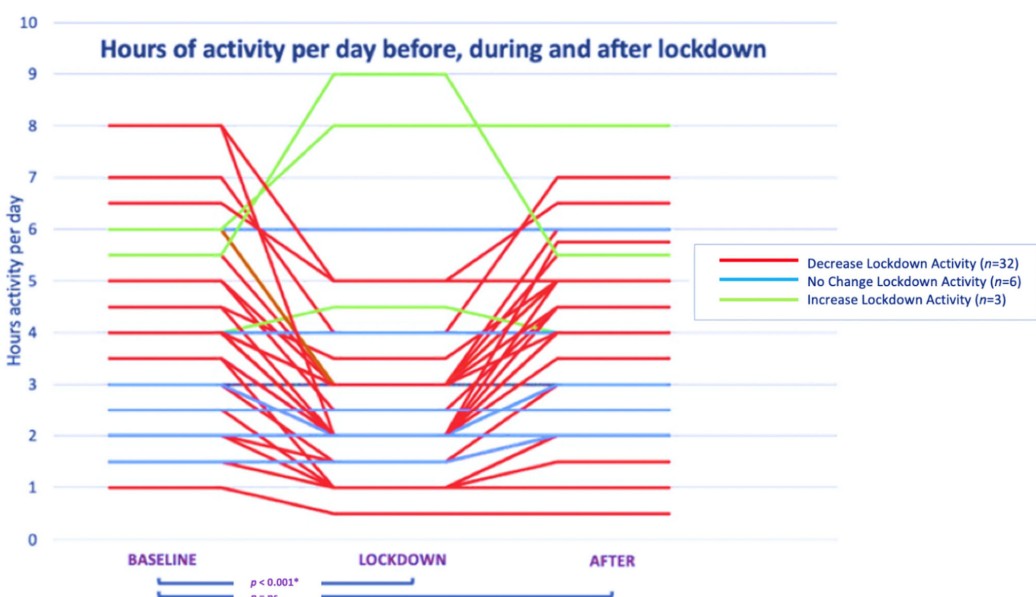

**Figure 1.** Hours of activity per day before, during, and after lockdown. * = indicates significance; ns = not significant.

Patients who had a decrease in their active hours (*n* = 32) were compared with those who were unchanged or had an increase in their activity level (*n* = 9). Patients who experienced a decrease in their activity were younger (mean age 27.9 ± 13.2 years) compared to the no change or increased activity group (35.3 ± 9.1 years) *p* = 0.03.

## 4. Discussion

Though several studies support the notion that physical activity decreased during the COVID-19 pandemic, the majority analyzes self-reported questionnaire or interview data. A decrease in physical activity and an increase in sedentary lifestyle were noted in children and adolescents (ages 5–17), young adults (ages 18–35), adults (ages 18–64) and the elderly (age > 65) [4–9]. More robust and reliable physical activity tracking methods are found sparsely in the literature. One study used daily step counts from smartphone applications to demonstrate a 27.3% decrease in mean steps within the first 30 days of the pandemic worldwide, but did not address the specific dates of regional lockdown nor how activity responded after lockdown restrictions were lifted [10].

It is well-understood that physical activity level is an important predictor of cardiovascular outcomes [3]. The decrease in physical activity during the COVID-19 pandemic has been linked to an associated increase in the global burden of cardiovascular disease [4]. Tracking physical activity data with implantable device accelerometers is optimal due to the implanted nature of the devices. As mentioned previously, some studies use smartphone accelerators to track metrics [10]. However, unlike smartphones, the present study ensures greater accuracy, as CRMs track patients' activity at all times. Another benefit of the study lies in its inclusion of both children and adults. Implanted cardiac rhythm devices allow physical activity tracking for children of all ages, irrespective of smartphone possession.

Patients in our study who experienced a decrease in physical activity were found to be younger than those who were unchanged or had an increase in their activity levels. This may be due to the nature of physical activity in children and adolescents, which often centers around group and team sports, compared with the physical activity behavior of adults, which is often a solo activity. The limitation of social interaction was the goal of the lockdown; however, the unintended consequence of decreased physical activity may have weighed more on younger individuals for this reason.

Limitations from this study mostly stem from the population analyzed. This study had a somewhat small sample size and was geographically focused on New York and New

Jersey. Additionally, participants were only included if they had accelerometer-enabled cardiac devices, which limited the sample size.

This study hoped to bring to the forefront that if there were to be another lockdown, the public should be encouraged to continue physical activity even while at home. Three study participants exhibited increases in physical activity from baseline, though this was not the norm. These three patients all acknowledged that they made a conscious effort to remain active and indeed made exercise a part of their lockdown regimen. In addition to the cardiovascular benefits, physical activity can help with mental health (depression, anxiety) and the prevention of several chronic diseases (diabetes, cancer, obesity, and osteoporosis) [12]. The present study demonstrates a clear role for public health encouragement of at-home physical activity in times of significant restriction in order to support all aspects of health beyond simple avoidance of virus infection and transmission.

**Author Contributions:** Conceptualization, R.F., D.J., J.R. and B.L.; methodology, R.F., D.J., J.R. and B.L.; formal analysis, R.F., D.J., J.R. and B.L.; writing—original draft preparation, R.F., D.J., J.R. and B.L.; writing—review and editing, R.F., D.J., J.R. and B.L.; supervision, B.L. All authors have read and agreed to the published version of the manuscript.

**Funding:** This research received no external funding.

**Institutional Review Board Statement:** The study was conducted in accordance with the Declaration of Helsinki, and approved by the Institutional Review Board of the Mount Sinai Health System, NY, USA (protocol code 21-01352 on 21 October 2021).

**Informed Consent Statement:** Patient consent was waived due to the deidentified nature of the data, impracticality of obtaining consent, and minimal risk to subjects.

**Data Availability Statement:** Not applicable.

**Conflicts of Interest:** The authors declare no conflict of interest.

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
