# Peer review of "Quantifying the Impact of COVID-19 “Lockdown” on Physical Activity in Children and Adults with Implanted Cardiac Rhythm Devices: A Single Center Experience Using Cardiac Rhythm Device Accelerometer Data"

_covid, doi:10.3390/covid2090089_

Round 1

Reviewer 1 Report

This is a very interesting communication coming to fill a void: whilst questionnaire based information has been examined before, objective analysis of activity during lock down is not equally common. This is a welcome contribution, even if it does not provide new surprising findings but rather a confirmation on what was already suspected.

The implications for public health policy and communication in the case of a future lockdown are important and clearly pointed out by the authors.

As a minor comment, in figure 1, when lines of the same color intersect, it is not possible to distinguish between and know which end connects to which start. Slightly different colors or multiple diagrams might enhance readability.

Author Response

Thank you for your comments.  We adjusted the colors, brightness and contrast of the image to improve readability.

Reviewer 2 Report

Manuscript ID: covid-1824768

Title: Quantifying the Impact of COVID-19 “lockdown” on physical activity in children and adults with implanted cardiac rhythm devices: A single center experience using cardiac rhythm device accelerometer data

Journal: COVID

Abstract

1) Lines 23-24. Instead of two different sets of paired-samples t-test, Authors should perform a repeated measures ANOVA with time as a within-subjects factor with three levels (before, during, and after the lockdown). In case of a significant effect, they should perform a Scheffé post-hoc test in order to detect which time points significantly differ from the others.

Introduction

1) Lines 45-47. Authors wrote that “However, the association between lockdown and physical activity remains poorly documented from an objective standpoint”. However, it should be acknowledged that previous smartphone-based studies (line 45) probably used the smartphone accelerometer.  

2) Are there any expected results?

3) Are there any similar-previous studies on the same clinical population?

Materials and Methods

1) Authors should provide more descriptive information about the sample. For example, are there any significant differences in gender distribution and age between the three sub-samples?

2) Line 63. Authors should provide more details about the accelerometer. For example, which is its sensitivity?

3) Line 66. The name of the company should be quoted. Moreover, validation studies of this algorithm should be quoted too.

4) Lines 72-73. Authors should refer to my comment number 1 on the abstract.

Results

1) Authors should report the complete statistics, not only the p-value.

Discussion

1) Lines 101-103. Authors wrote “One study used daily step counts from smart phones applications to demonstrate a 27.3% decrease in mean steps within the first 30 days of the pandemic worldwide [10]”. However, that study was not focused on the clinical population investigated in the current work.

2) Lines 119-120. I would suggest that Authors remove “therefore making the results more generalizable to the general population” because, anyway, they investigated a clinical sample.

3) Lines 124-125. Authors wrote “These three patients all acknowledged that they made a conscious effort  to remain active and indeed made excercise a part of their lockdown regimen”. I was wondering whether the cardiovascular status of those three patients was different at the end of lockdown compared to the others.

Informed consent statement

1) Authors wrote “Patient consent was waived due to the deidentified nature of the data, impracticality of obtaining consent, and minimal risk to subjects”. Was the patient consent waived by the Institutional Review Board of the Mount Sinai Health System (protocol code 21-01352 on October 21, 2021)?

Author Response

Thank you for taking the time to thoughtfully comment on the manuscript.   As you will see in the response, asking for additional analysis uncovered some interesting new findings!

1) Lines 23-24. Instead of two different sets of paired-samples t-test, Authors should perform a repeated measures ANOVA with time as a within-subjects factor with three levels (before, during, and after the lockdown). In case of a significant effect, they should perform a Scheffé post-hoc test in order to detect which time points significantly differ from the others.

Our statistician feels that the paired T test is sufficient to use for comparisons here.  There are only 3 time points, not multiple, so the added power of the ANOVA was not felt to be necessary. Though there may be advantages to this more powerful statistical method, redoing the analysis will not be possible for us in the revision time-frame.

1) Lines 45-47. Authors wrote that “However, the association between lockdown and physical activity remains poorly documented from an objective standpoint”. However, it should be acknowledged that previous smartphone-based studies (line 45) probably used the smartphone accelerometer. 

Agreed. Will change the wording to “incompletely characterized”.  We go on to discuss the limitations of smartphone-based data in the discussion.

2) Are there any expected results?

This work was exploratory.  Indeed until we saw the patterns emerge we didn’t know what we were going to find.

3) Are there any similar-previous studies on the same clinical population?

 No which is why this has been incompletely characterized.

Materials and Methods

1) Authors should provide more descriptive information about the sample. For example, are there any significant differences in gender distribution and age between the three sub-samples?

Thank you for the suggestion.  We found something interesting.  The majority of the patients with decreased activity (n=32) had a mean age of 27.9 +/- 13.2 years whereas the patients with no change or increased activity were older 35.3+/- 9.1 years.  P=0.03.  (We grouped the no change and increased groups together as there were only 3 patients in the increased group and the mean of those patients was exactly the same as the no change group)

We included a discussion of this finding.  We suspect that children and young adults do much of their physical activity in structured group activities whereas adults often do much of their physical activity as a solitary activity.  Solitary activities are much easier to keep up during lockdown. Unfortunately our data set does not allow us to explore this further but has sparked an idea for future research endevor!

2) Line 63. Authors should provide more details about the accelerometer. For example, which is its sensitivity?

We provide a brief description of the general function of the accelerometer however the exact functioning is proprietary to the devices. The devices are all Medtronic and Boston Scientific.   However the general principle and validation has been described (reference 11). 

3) Line 66. The name of the company should be quoted. Moreover, validation studies of this algorithm should be quoted too.

See response above.  We included a line about the manufacturers.

4) Lines 72-73. Authors should refer to my comment number 1 on the abstract.

Addressed above

Results

1) Authors should report the complete statistics, not only the p-value.

We report the mean, standard deviation and P valve for T test

Discussion

1) Lines 101-103. Authors wrote “One study used daily step counts from smart phones applications to demonstrate a 27.3% decrease in mean steps within the first 30 days of the pandemic worldwide [10]”. However, that study was not focused on the clinical population investigated in the current work.

Agreed however both studies discuss physical activity during the pandemic which is relevant.

2) Lines 119-120. I would suggest that Authors remove “therefore making the results more generalizable to the general population” because, anyway, they investigated a clinical sample.

Removed that line.

3) Lines 124-125. Authors wrote “These three patients all acknowledged that they made a conscious effort to remain active and indeed made excercise a part of their lockdown regimen”. I was wondering whether the cardiovascular status of those three patients was different at the end of lockdown compared to the others.

Unfortunately not possible to assess with the limitations of the study because this was observational after-the-fact.

Informed consent statement

1) Authors wrote “Patient consent was waived due to the deidentified nature of the data, impracticality of obtaining consent, and minimal risk to subjects”. Was the patient consent waived by the Institutional Review Board of the Mount Sinai Health System (protocol code 21-01352 on October 21, 2021)?

Yes.  We obtained the waiver of consent from the Mount Sinai IRB with the above protocol and date.

Reviewer 3 Report

Rebecca Fisher et al examined the effect of the public health lockdown on physical activity in a group of patients with cardiac rhythm devices. They found that the COVID 19 lockdown in NY/NJ during the Spring of 2020 resulted in a significant drop in active hours/day in children and adults with congenital heart disease. Active hours/day rebounded to baseline after restrictions were lifted.

The manuscript is well written, the methods are clear and the conclusions are comprehensive.

Line 54: Typo "physical"

Author Response

Thank you for your comments

Typo corrected.

Reviewer 4 Report

The analyzed volume of patients is not large, which is also indicated by the authors. Another minor limitation is one geographical region. The third limitation is a single load adaptation system. At the same time, these limitations do not significantly lower the value of the work, because in my opinion, the conducted research is very important, analyzing high-risk patient groups with a very accurate and objective methodology.

Author Response

Thank you for your comments.  We agree with the limitations you pointed out.

Round 2

Reviewer 2 Report

Manuscript ID: covid-1824768

Title: Quantifying the Impact of COVID-19 “lockdown” on physical activity in children and adults with implanted cardiac rhythm devices: A single center experience using cardiac rhythm device accelerometer data

Journal: COVID

Authors’ reply to my previous comment number 1 on the Abstract

1) I still believe that performing a repeated measures ANOVA with time as a within-subjects factor with three levels (before, during, and after the lockdown) is more appropriate than performing two different sets of paired-samples t-test. Moreover, performing the new statistical analysis should not be so time consuming.

Authors’ reply to my previous comment number 1 on the Results

1) Authors should also report the degrees of freedom and the t-values.

Author Response

1) I still believe that performing a repeated measures ANOVA with time as a within-subjects factor with three levels (before, during, and after the lockdown) is more appropriate than performing two different sets of paired-samples t-test. Moreover, performing the new statistical analysis should not be so time consuming.

We again asked our statistical support team who helped us with this manuscript and they feel the T test is appropriate.  They would agree to do a repeated measures ANOVA for us, but would require resubmission of a new request that would probably take at least 8 weeks to get redone.  If the reviewer feels this is a deal-breaker then we will have to submit someplace else.

1) Authors should also report the degrees of freedom and the t-values.

DF and T values included.